# Iron Nutrition in Plants: Towards a New Paradigm?

**DOI:** 10.3390/plants12020384

**Published:** 2023-01-13

**Authors:** Meijie Li, Shunsuke Watanabe, Fei Gao, Christian Dubos

**Affiliations:** 1IPSiM, University Montpellier, CNRS, INRAE, Institut Agro, Montpellier, France; 2College of Agronomy, Hunan Agricultural University, Changsha 410128, China

**Keywords:** Arabidopsis, bHLH, dicots, grass, iron homeostasis, transcription factor

## Abstract

Iron (Fe) is an essential micronutrient for plant growth and development. Fe availability affects crops’ productivity and the quality of their derived products and thus human nutrition. Fe is poorly available for plant use since it is mostly present in soils in the form of insoluble oxides/hydroxides, especially at neutral to alkaline pH. How plants cope with low-Fe conditions and acquire Fe from soil has been investigated for decades. Pioneering work highlighted that plants have evolved two different strategies to mine Fe from soils, the so-called Strategy I (Fe reduction strategy) and Strategy II (Fe chelation strategy). Strategy I is employed by non-grass species whereas graminaceous plants utilize Strategy II. Recently, it has emerged that these two strategies are not fully exclusive and that the mechanism used by plants for Fe uptake is directly shaped by the characteristics of the soil on which they grow (e.g., pH, oxygen concentration). In this review, recent findings on plant Fe uptake and the regulation of this process will be summarized and their impact on our understanding of plant Fe nutrition will be discussed.

## 1. Introduction

Iron (Fe) is an essential micronutrient for almost all living organisms since it is found in cofactors ensuring the activity and/or the stability of metalloproteins involved in a plethora of physiological processes (e.g., respiration, photosynthesis, sulfur and nitrogen assimilation, amino acid biosynthesis) [1,2]. For most organisms, perturbations in Fe homeostasis led to cellular defects that negatively impact growth.

In humans, Fe homeostasis disorders are associated with several health issues such as cancer risk, neurodegenerative diseases, and Fe deficiency anaemia. Fe deficiency anaemia is the most prominent since it affects about one billion people worldwide [3]. Improving Fe content in their diet would allow overcoming the associated symptoms. Indeed, for several reasons (feasibility, sustainability, and cost), increasing meat consumption is not an issue even if humans preferentially absorb haem-bound Fe present in large amounts in animal flesh. Therefore, improving Fe content and bioavailability in crops through biofortification would have a tremendous beneficial effect on human health. A key step in achieving this goal is to decrypt the molecular mechanisms that regulate Fe homeostasis in plants.

Several studies have shown that in plants, Fe excess leads to plant growth defects and crop yield decrease [4]. This is because in aerobic conditions Fe can react with hydrogen peroxide (H_2_O_2_) generating reactive oxygen species (ROS), via the so-called Fenton reaction, which is deleterious to the cell [5]. Nonetheless, Fe deficiency is widely spread even if Fe is the fourth most abundant element on Earth. This is because Fe is mostly present in soils in the form of poorly soluble oxides/hydroxides, especially at neutral to alkaline pH [6,7]. Thus, Fe homeostasis in plant cells must be tightly regulated to avoid Fe deficiency or Fe excess, both of which severely impact the yield of crops and the quality of their derived products [8].

To this end, plants have evolved several molecular mechanisms to modulate the levels of Fe present in cells. Within the past decades, tremendous efforts have been made by the scientific community to elucidate the nature and the components of these mechanisms. A landmark in these studies was the description of two different Fe uptake strategies discriminating grass and non-grass species [9,10], the so-called Strategy I and Strategy II.

In this review, recent findings on plant Fe uptake and the regulation of this process will be summarized and their impact on our understanding of plant Fe nutrition will be discussed.

## 2. Strategy I

Strategy I (reduction-based strategy) is mostly used by dicots and non-graminous monocots. Strategy I relies on the combined activity of three types of protein localized at the plasma membrane of the root epidermis whose expression is induced when Fe availability is low (Figure 1a) [11]. First, P-type ATPases secrete protons into the rhizosphere reducing the pH and therefore increasing the solubility of Fe. It is noteworthy that the excretion of solubilizing agents, such as carboxylates and phenolates, participates also in this process. Then, FERRIC REDUCTION OXIDASES, whose activity is promoted at acidic pH [12], reduce Fe^3+^ into Fe^2+^. Last, ZIP (ZINC-REGULATED TRANSPORTER/IRON-REGULATED TRANSPORTER-LIKE PROTEIN) transporters take up Fe^2+^ into the plant roots. In Arabidopsis, these three proteins are encoded by *AHA2* (*H^+^-ATPase 2*), *FRO2* (*FERRIC REDUCTION OXIDASES 2*) and *IRT1* (*IRON-REGULATED TRANSPORTER 1*), respectively [11]. It is noteworthy that IRT1 is a high-affinity Fe transporter that is crucial for Fe uptake and root-to-shoot Fe partitioning [13,14]. In addition to IRT1, NRAMP1 (NATURAL RESISTANCE-ASSOCIATED MACROPHAGE PROTEIN 1) participates in Fe uptake as a low-affinity Fe^2+^ transporter [15,16]. It was recently demonstrated that AHA2, FRO2 and IRT1 form a complex intended to create a local environment of pH and Fe^2+^ concentration in the rhizosphere, which optimizes Fe uptake by avoiding Fe^2+^ reoxidation into Fe^3+^ due to the oxygen that is present in most soils [17].

Strategy I was initially thought to be restricted to dicots and non-graminous monocots but several studies showed that grass species also use this strategy to take up Fe. For instance, rice (*Oryza sativa*), a grass species, takes up Fe^2+^ via the activity of OsIRT1, OsIRT2, OsNRAMP1 and OsNRAMP5 (Figure 1a) [18,19,20]. The expression of *OsIRT1*, *OsIRT2* and *OsNRAMP1* is induced in response to Fe deficiency indicating that these transporters play a role in plant Fe nutrition when Fe availability is low [19]. In contrast, OsNRAMP5, whose expression is not induced in response to Fe deficiency, ensures constitutive Fe^2+^ uptake [18]. Interestingly, under flooded anaerobic conditions, when Fe is mostly present in the form of Fe^2+^, *OsIRT1* expression is downregulated whereas that of *OsIRT2* is induced [21] Such as the non-grass species, grasses are also able to reduce the Fe^3+^ present in the rhizosphere into Fe^2+^. Instead of releasing protons and enzymatically reducing Fe^3+^, grasses secrete phenolic compounds such as protocatechuic and caffeic acids. In rice, these steps are achieved by the PHENOLICS EFFLUX ZERO 2 (PEZ2) transporter [18].

## 3. A Current Understanding of Strategy II

Contrary to Strategy I, Strategy II (Fe chelation strategy) is predominantly employed by graminaceous monocots including industrially valuable crops such as rice. Fe acquisition driven by Strategy II has intensively received attention for a long time toward improvements in agricultural yield. Phytosiderophores (PS) are organic compounds which function as naturally occurring metal chelators and play an essential role in Strategy II. Since the compounds are capable of a multidentate chelate of Fe, Strategy II-type plants massively secrete them from roots into the rhizosphere by a specific transport system for the mobilization of inaccessible Fe in soil clay particles. Strategy II is regarded to be a more high-efficient Fe acquisition system than Strategy I, allowing graminaceous plants to overcome more harsh Fe deficiency like alkaline soils.

Until now, mugineic acids (MA), avenic acids, distichonic acids and nicotianamines (NA) had been reported as PS families in Strategy II-type plants (Figure 1b). In particular, the MA family, consistent with 2′-deoxymugineic acid (DMA), and its derivatives is well known as the major PS cluster, which was originally identified in root washings of rice, oats and barley [22,23]). The MA family compounds (MAs) are classified into non-proteinogenic amino acids and carry tricarboxylic groups, enabling them to generate coordination bonds with Fe^3+^. In the MAs biosynthesis pathway, DMA is the initial compound synthesized from S-adenosyl-l methionine through two catalytic reactions mediated by NICOTIANAMINE SYNTHASE (NAS) and NA AMINOTRANSFERASE (NAAT). DMA is then further converted to its hydroxylation derivatives [MA, 3-hydroxymugineic acid, 3-epihydroxy-2′-deoxymugineic acid, 3-epihydroxymugineic acid (epiHMA)] [24,25]. A large number of MAs produced in response to Fe deficiency is subsequently secreted from roots into the rhizosphere by the TOM1 (TRANSPORTER OF MA 1) PS efflux transporter [26,27]. Root-secreted MAs then couple with Fe^3+^ in the vicinity of the roots, resulting in the formation of Fe-MA complexes. Finally, the complexes are absorbed into root cells via the YS1 (YELLOW STRIPE1) and YSL (YS1-LIKE) transporters.

MAs have a wide range of variety in Fe-chelating activity and Fe-MA-complex stability depends on their structures [28]. For instance, hydroxylated MA species including MA and epiHMA are reported to be more stable under weak acid conditions than their non-hydroxylated forms, which could influence the Fe-uptake ability of plants [28]. The diverse MA properties are thought to be an advantage in Strategy II for acclimating to Fe deficiency caused by multiple environmental factors such as aeration conditions, redox status, and pH of the soil. In addition, the various properties of MAs might also play an important role in the distribution of Fe within the plant because Fe is translocated as a Fe-chelator complex through the xylem and phloem whose pH is weak acid and alkaline, respectively [24,29].

The chelation strategy is considered specific to grass species; however, recent studies suggest that this is likely not to be the case and that non-grass species use a Strategy II-like to take up Fe. For instance, it was shown that Arabidopsis secretes from roots to the rhizosphere Fe-mobilizing coumarins (FMC) for promoting Fe uptake, a mechanism closely similar to that of the MA-driven Strategy II (Figure 1b) [30].

Esculetin, fraxetin and sideretin belonging to the catechol coumarin group are major FMCs detected in root secretion of Arabidopsis under Fe deficiency. The catechol moiety of these FMCs is an essential structure to interact with and mobilize insoluble Fe in the rhizosphere. These compounds are synthesized from feruloyl-coenzyme A (CoA), which is an intermediate of the phenylpropanoid pathway, via esculetin or scopoletin [31]. The FERULOYL-CoA 6′HYDROXYLASE 1 (F6′H1) catalyses the first reaction in the FMC biosynthesis, leading to the production of 6′-hydroxyferuloyl-CoA. The conversion of 6′-hydroxyferuloyl-CoA into scopoletin is mainly catalysed by COUMARIN SYNTHASE (COSY), followed by the hydroxylation of scopoletin by the SCOPOLETIN 8-HYDROXYLASE (S8H) for producing fraxetin. CYP82C4, a Fe deficiency-responsive cytochrome P450 enzyme, convert fraxetin into sideretin. Esculetin biosynthesis is still an open question. Once synthesized, FMCs are either secreted in their aglycon form or stored in the vacuole in their glycosylated form. UDP-GLYCOSYLTRANSFERASE (UGT) encoded by UGT72E cluster genes mediates this later step [32]. Stored FMCs are released from their glycosides by β-glucosidases such as BGLU42 prior to their secretion into the rhizosphere via the PDR9/ABCG37 (PLEIOTROPIC DRUG RESISTANCE 9/ATP-BINDING CASSETTE G SUBFAMILY 37) transporter [33,34]).

It was initially assumed that FMCs facilitate the mobilization of insoluble Fe in the soil in coordination with proton extrusion. More recent findings, however, suggest that fraxetin is capable of directly binding with Fe^3+^ and forming Fe-fraxetin complexes in alkaline conditions in a manner similar to that of MAs [30]. Given that fraxetin secretion is more conspicuous under alkaline conditions in which ferric-chelate reductase activity is impaired [12], Strategy II-like Fe acquisition seems a highly plausible system for non-graminaceous plants. Fraxetin may be preferable at high pH conditions because of its potentiated stability and ability to stably chelate Fe^3+^ even though sideretin is the major FMC in root secretion both under low and high pH Fe deficient conditions [35].

An uptake mechanism of Fe-fraxetin complexes remains largely obscure, except for some pieces of information provided by pharmacological studies [30,36]; (i) FMCs are uptaken by the root cells; (ii) the supplementation of fraxetin can rescue Fe starvation phenotypes of mutants defective in coumarin biosynthesis in alkaline conditions; (iii) the recovery effects of fraxetin in Fe acquisition is abolished by supplying inhibitors of ATP-dependent transport activities. This experimental evidence strongly suggests the participation of a specific transport system in Fe-fraxetin uptake in roots that could be widely conserved within non-graminaceous species. The question of how plants absorb Fe-fraxetin complexes will need to be answered in future studies.

One can consider that Strategy II and the emerging Strategy II-like are totally different mechanisms since the Fe-mobilizing compounds, Fe-mobilizing compound biosynthesis sites, enzymes, and transporters, which participate in each strategy, are not conserved [11,24,31]. Nevertheless, from the perspective of the use of root-born metabolites possessing Fe chelation activity for mobilizing Fe, Strategy II using MAs and the Strategy II-like strategy using FMCs are inclusive in their fundamental principles rather than exclusive. These similarities suggest a convergent evolution of the two chelation-based Fe uptake strategies by different environmental pressures as a driving force, but toward the same direction.

Recent understanding regarding the physiological functions of root exudates is raising the possibility of no definite border between Fe acquisition systems in Strategy I and Strategy II plants. In fact, unlike Fe uptake, long-distance transport systems of Fe are partially shared between these plants. NAS orthologs are widely conserved in plant kingdoms, and they have the ability to produce NA, while NAAT is specifically developed in Strategy II-type plants [37]. In Arabidopsis, NA chelates Fe^2+^ present in apoplastic space instead of MAs, leading to the formation of Fe-NA complexes, which is a major substrate for long-distance Fe transport through the phloem. Fe-NA loading into the phloem is mediated by YSL1 and YSL3, orthologs of OsYSL18. NA EFFLUX TRANSPORTERs (NAET1 and NAET2) were recently reported to be responsible for NA secretion into apoplastic space in Arabidopsis [38].

## 4. Transcriptional Regulation of Fe Homeostasis

As described above, the comparison of the two Fe uptake strategies highlighted that the molecular mechanisms employed are after all quite similar whether plants belong to grass or non-grass species. Instead, environmental parameters, in particular soil properties (i.e., pH, oxygen content), seem to be more relevant to predict which strategy plants may preferentially use.

How Fe uptake is regulated at the molecular level has also been investigated for decades, pinpointing the preponderant role played by the transcriptional machinery [11,39,40,41,42,43]. If these studies have led to the identification of transcription factors belonging to several families (e.g., ABI3/VP1, ARF, bZIP, B3, C2H2, EIL, ERF, MYB, NAC, NFY, WRKY, YABBY; [11]; Table 1), it has emerged that the backbone of the transcriptional regulatory cascade controlling Fe homeostasis in plants, and thus Fe uptake, relies on the activity of several bHLH proteins whose activity is conserved between grass and non-grass species. The first transcription factor identified as directly involved in the regulation of Fe uptake was the tomato (*Solanum lycopersicum* L.) bHLH FER [44]. Since the characterization of FER, 17 different bHLH proteins (from six bHLH subfamilies) were shown to participate in the maintenance of Fe homeostasis in Arabidopsis. Indeed, functional homologues for several of these genes have been characterized in rice. These bHLH transcription factors are interacting in an intricate regulatory network composed of two interconnected modules (Figure 2).

In Arabidopsis, the first module relies on FIT/bHLH29 (FER-LIKE IRON DEFICIENCY INDUCED TRANSCRIPTION FACTOR) which is the functional homolog of FER [45,46,47]. FIT belongs to the bHLH clade IIIa and is a direct positive regulator of *IRT1* and *FRO2* expression [48]. To achieve its function, FIT interacts with the four bHLH members of the clade Ib (i.e., bHLH38, bHLH39, bHLH100 and bHLH101) to form heterodimers with partially overlapping functions (Figure 2). In rice, OsFIT/OsbHLH156 is a positive regulator of genes encoding proteins involved in the biosynthesis of MAs and the uptake of Fe-MA complexes (e.g., *OsNAS1*, *OsNAS2*, *OsYSL15*) as well as *OsIRT1* [49,50]. OsIRO2/OsbHLH56 is a clade Ib bHLH that, like its Arabidopsis homologs, acts as a positive regulator of Fe uptake and interacts with OsFIT [51,52]. Interestingly, it was also shown that the FIT-dependent nuclear localisation of clade Ib bHLH is also conserved in grass and non-grass species supporting the conservation of FIT/clade Ib heterodimers formation between Strategy I and Strategy II plants [49,50,53]). However, unlike what is described in Arabidopsis, OsIRO2 is not involved in the regulation of *OsIRT1* expression. Whether the regulation of *OsIRT1* expression by OsFIT involves Ib bHLH transcription factors other than OsIRO2 or differs from the homologous Arabidopsis regulatory system is still to be determined.

The second module acts upstream from FIT and involves bHLH transcription factors from clade IVc. The four Arabidopsis members are ILR3/bHLH105 (IAA-LEUCINE RESISTANT 3), IDT1/bHLH34 (IRON DEFICIENCY TOLERANT 1), bHLH104 and bHLH115. Their role in the regulation of Fe uptake and Fe homeostasis relies on their ability to form homo and heterodimers [54,55,56]; Figure 2). Among their target genes are clade Ib bHLH transcription factors (direct) and *FIT* (indirect). Similarly, the four rice clades IVc bHLH are the positive regulator of the Fe deficiency response [41,57,58,59], namely OsPRI1/OsbHLH60, OsPRI2/bHLH58, OsPRI3/OsbHLH59 (POSITIVE REGULATOR OF IRON HOMEOSTASIS 1, 2 and 3) and OsbHLH57. Like their Arabidopsis homologs, PRI proteins directly target clade Ib bHLH (i.e., *OsIRO2*).

Clade IVb bHLH transcription factors play also a central role in the regulation of Fe homeostasis and therefore Fe uptake. Based on their function and amino acid sequence features, clade IVb bHLH can be divided into two subgroups. The first subgroup is composed of the Arabidopsis PYE/bHLH47 and the rice OsIRO3/bHLH63 and OsIRO4/bHLH61 (Figure 2). Both proteins contain a transcriptional repressive EAR motif [60] in their C-terminal region and negatively regulate the expression of genes involved in the maintenance of Fe homeostasis [61,62,63]. PYE, OsIRO3, and OsIRO4 also interact with the clade IVc bHLH ILR3 and bHLH115, and OsPRI1 and OsPRI2, respectively. The role of these interactions was still not understood until recently when it was found that ILR3 play a central role in the regulation of Fe homeostasis, where it acts as both a transcriptional activator of the plant responses to Fe deficiency and a repressor of the responses to Fe excess [62]. In this study, it was found that when ILR3 interacts with clade IVc bHLH it acts as a transcriptional activator of genes involved in Fe uptake (e.g., clade Ib bHLH genes) and when it interacts with PYE it acts as a transcriptional repressor of genes involved in Fe storage (e.g., *FER1*, *FER3* and *FER4* ferritin genes). Interestingly, among the direct target of PYE-ILR3 complexes, there is *PYE* itself, forming a negative feedback regulatory loop when Fe availability is not limiting [62]; Figure 2). Such a regulatory loop would allow balancing Fe uptake and Fe storage to meet the Fe demand of the plant for its growth and development. Similarly to PYE, the interaction of OsIRO3 and OsIRO4 with OsPRI1 or OsPRI2 leads to the inhibition of the expression of their target genes, including *OsIRO3* itself [61,63]. Interestingly, unlike PYE, no gene involved in Fe storage has been identified to date as a direct target of OsIRO3 and OsIRO4. As expected, the repressive activity of OsIRO3 and OsIRO4 conferred by their EAR domain occurs via the recruitment of TOPLESS/TOPLESS RELATED repressors [60,63,64]. Whether this is also the case for PYE is still unclear and remains to be elucidated. In Arabidopsis, the second subgroup of clade IVb bHLH is composed of bHLH11 and URI/bLHL121 (UPSTREAM REGULATOR OF IRT1). *bHLH11* encodes, like *PYE* and *OsIRO3*, a transcriptional repressor [65]. bHLH11 interaction with clade IVc bHLH [61,66] leads to the inhibition of both FIT function (indirect) and the expression of clade Ib bHLH genes by recruiting TOPLESS/TOPLESS RELATED repressors via its two EAR motifs [61,65]. The role of bHLH11 would be to prevent Fe over-accumulation and thus toxicity when Fe availability is high. In contrast, URI is a transcriptional activator that can form heterodimers with clade IVc bHLH to induce the Fe uptake machinery by activating the expression of *FIT* (indirect) and clade Ib bHLH genes (direct) [66,67,68,69]. Interestingly, the cellular localization of URI in roots, and not its transcript and protein accumulation, differs depending on the availability of Fe [66,67]. Under Fe-sufficient conditions, URI mainly localizes in the stele and the endodermis, whereas under Fe deficiency URI is primarily observed in the cortex and the epidermis cells, where it promotes Fe uptake [66]. The in-depth characterization of URI highlighted that it plays a key role in the control of plant Fe homeostasis since it directly or indirectly regulates the expression of most of the known genes involved in this intricate transcriptional regulatory network [66,67]. In contrast to PYE (and ILR3), URI directly activates the expression of the three main ferritin genes (i.e., *FER1*, *FER3* and *FER4*) when Fe is not in excess, indicating that URI maintains Fe homeostasis by positively regulating the transient storage of Fe as well as the Fe deficiency response [70]. This later observation poses the question of whether URI regulates the expression of some of its target genes independently of clade IVc bHLH transcription factors. Surprisingly, to date, no functional homologs have been characterized in grass species for bHLH11 and URI.

## 5. Post-Translational Regulation of the Fe Homeostasis Regulatory Network, Fe Sensing and Long Distance Signalling of Fe Status

Post-translational modifications are efficient mechanisms to modulate the activity of transcription factors to adapt Fe uptake to plant needs [71,72,73]. Among them, ubiquitination has emerged as an important mechanism to regulate the stability, and thus the amount and activity, of key transcription factors involved in the regulation of Fe uptake in plants. This mechanism is conserved between grass and non-grass species and relies on a set of specific E3-ubiquitin ligases whose activity prevents plants from potential Fe overload. These proteins contain three hemerythrin motifs at the N-terminus side and a RING-type E3 ubiquitin ligase at the C-terminal end [72]. The hemerythrin domains bind Fe and contribute to the instability of the protein that is degraded via the 26S proteasome pathway following self-ubiquitination or ubiquitination by other E3 ligases [59,74,75].

The first hemerythrin E3-ubiquitin ligase involved in the regulation of Fe homeostasis was identified in Arabidopsis and was named BTS (BRUTUS) [76]. Rice functional homologs were also characterized and named HRZ1 and 2 (HAEMERYTHRIN MOTIF-CONTAINING REALLY INTERESTING NEW GENE (RING) AND ZINC-FINGER PROTEIN 1 and 2) [74].

*BTS*, *HRZ1* and *HRZ2* expression is induced in response to Fe deficiency, particularly in shoots and loss-of-function mutants over-accumulate Fe [74,77,78]. The encoded proteins interact with clade IVc bHLH (i.e., ILR3 and bHLH115 for BTS and OsPRI1 and OsPRI2 for HRZ1 and HRZ2) facilitating their degradation via the 26S proteasome pathway ([41,58,59,75,76]; Figure 2). The hemerythrin E3-ubiquitin ligase-dependent degradation of clade IVc bHLH allows fine-tuning of the expression of downstream Fe deficiency response genes and thus Fe uptake.

If one considers, as proposed by [79], that a Fe sensor in a living system is a biomolecule that binds Fe, thereby changing its function to regulate Fe homeostasis, hemerythrin E3-ubiquitin ligases emerge as obvious candidates for both grass and non-grass species.

Interestingly, dicot species possess two additional hemerythrin E3-ubiquitin ligases localized in roots, namely BTSL1 and BTSL2 (BTS-LIKE 1 and 2), that specifically target *FIT*, which directly regulate the expression of *IRT1* and *FRO2* and thus Fe uptake [72]. The absence of such E3-ligase in the grass might reflect the minor role played by OsIRT1 for the uptake of Fe in these plant species.

Recently, a family of peptides has emerged as a positive regulator of Fe uptake genes in both grass and non-grass species [80,81,82,83]. These peptides were named IMA/FEP (IRONMAN/ FE-UPTAKE-INDUCING PEPTIDE). IMA/FEP peptides preferentially accumulate in the vasculature region and participate in the inter-organ signalling of cellular Fe status for fine-tuning Fe uptake in roots [81,84,85,86]. How IMA/FEP peptides are modulating the plant response to Fe availability was a matter of debate. Recent studies have shown that IMA/FEP peptides interact with the hemerythrin E3-ubiquitin ligases (i.e., BTS, HRZ1 and HRZ2) to inhibit the degradation of clade IVc bHLH and thus promote Fe uptake and translocation ([85,87,88]; Figure 2). It is proposed that IMA/FEP peptides compete with clade IVc bHLH as a substrate of the hemerythrin E3-ubiquitin ligases, leading to their ubiquitination and subsequent degradation by the 26S proteasome pathway [87]. Interestingly, no lysine has been reported in the predicted amino acid sequence of the Arabidopsis IMA3/FEP1 peptide [81,82] shown to be ubiquitinated by BTS [87]. This suggests that ubiquitination of IMA3/FEP1 peptides might occur through other amino acids than lysine, a mechanism not yet described in plants.

Altogether, these recent findings highlight that the post-translational regulation of clade IVc bHLH transcription factors by the combined action of hemerythrin E3-ubiquitin ligases and IMA/FEP peptides play a key role in the regulation of Fe uptake and partitioning between the different organs of the plants. Such regulatory mechanism relies on a feed-forward regulatory loop where IMA/FEP peptides modulate the hemerythrin E3-ubiquitin ligase-dependent degradation via the 26S proteasome pathway of clade IVc bHLH transcription factors whose activity directly controls the expression of *IMA/FEP* genes.

## 6. Conclusions

The comparison of grass and non-grass Fe-uptake machinery shows that the border initially described between Strategy I (reduction) and Strategy II (chelation) is blurring. Instead, it suggests that plants select Strategy I and Strategy II modules to adapt to specific environmental conditions. In support of this assertion, it was reported that wild dicots adapted to alkaline grassland use an alternate Strategy I for Fe uptake that resembles one of the rice plants [89]. Here, the acidification of the rhizosphere and the Fe^3+^ reducing activity of FRO proteins are inhibited because of the buffering capacity of carbonate/bicarbonate present in alkaline soils and because of the high pH, respectively [12,90]. Instead, these plants secrete in the rhizosphere metabolites (e.g., phenolic compounds) with Fe^3+^ reducing activities to take up Fe^2+^. Surprisingly, unlike what is usually observed for Strategy I plants, this root Fe^3+^ reducing activity drastically increases with the concentration of Fe present in the media [89].

Similarly, the core elements of the transcriptional network that regulate Strategy I and Strategy II Fe uptake machinery also display strong similarities. On this basis, it is likely, even if it remains to be demonstrated, that other levels of regulation might also be similar between both types of plants. Among them, one could cite the involvement of alternative splicing [91,92], miRNA- and Lnc-RNA-dependent regulation of gene expression [93,94,95], chromatin modifications (i.e., histone methylation/acetylation and DNA methylation; [96], as well as Fe-dependent mitogen-activated protein kinases (MPKs) signalling cascade [97] or the phosphorylation-dependent regulation of key proteins, as it is the case for IRT1, AHA2, FIT or URI [98].

It is noteworthy that the activities of the different Fe uptake strategies developed by plants to manage Fe homeostasis are themselves modulated by important soil characteristics other than pH, such as for instance humic acids that are thought to be part of the cross-talk between plants and soil [99].

Future studies will help to fully dissect the conserved Fe uptake and regulation mechanisms between grass and non-grass, which might offer novel opportunities for the breeding of Fe-biofortified crops.

## Figures and Tables

**Figure 1 plants-12-00384-f001:**
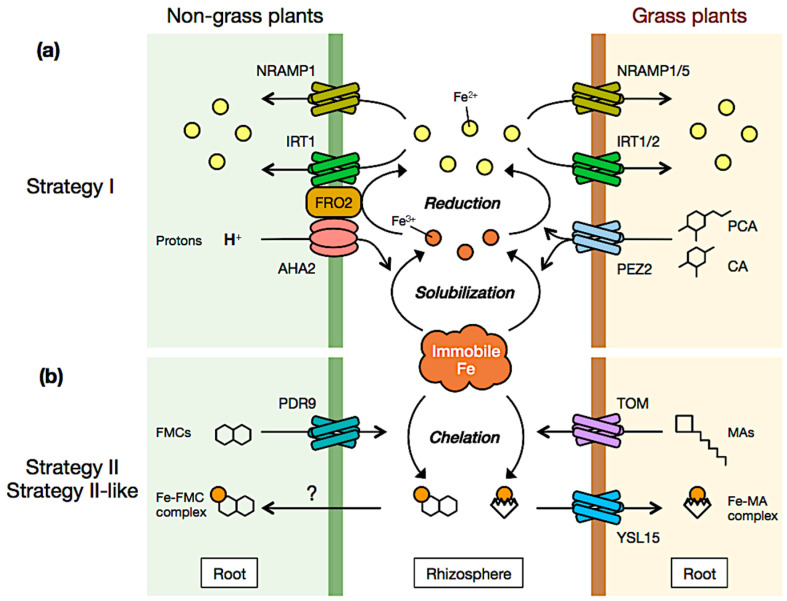
Iron uptake strategies in plants, not so different after all. (**a**) Strategy I, also called the reduction-based strategy, was first described as specific to non-grass species. In Arabidopsis, AHA2 (H^+^-ATPase) secretes protons into the rhizosphere to solubilise Fe^3+^ which is then reduced into Fe^2+^ by the ferric reductase FRO2. Fe^2+^ is then uptaken into the plant root via IRT1 and to a lesser extent NRAMP1. Recent findings highlighted that the solubilization/reduction steps can be achieved in rice via the secretion of protocatechuic (PCA) and catechuic (CA) acids via the PEZ2 transporter. Fe^2+^ uptake into the root is then insured by IRT1, IRT2, NRAMP1 and NRAMP5 activities. (**b**) Strategy II, also called the chelation strategy, was first described as specific to grass species. In rice, mugineic acids (MAs) are secreted via the TOM transporter and Fe^3+^-MA complexes are taken up into the root via the YSL15 transporter. In Arabidopsis, it has recently been shown that Fe mobilizing coumarins (FMC) can chelate Fe^3+^ following their secretion into the rhizosphere via the PDR9 transporter and that Fe^3+^-FMC complexes are taken up into the plant root via an ATP-dependent mechanism.

**Figure 2 plants-12-00384-f002:**
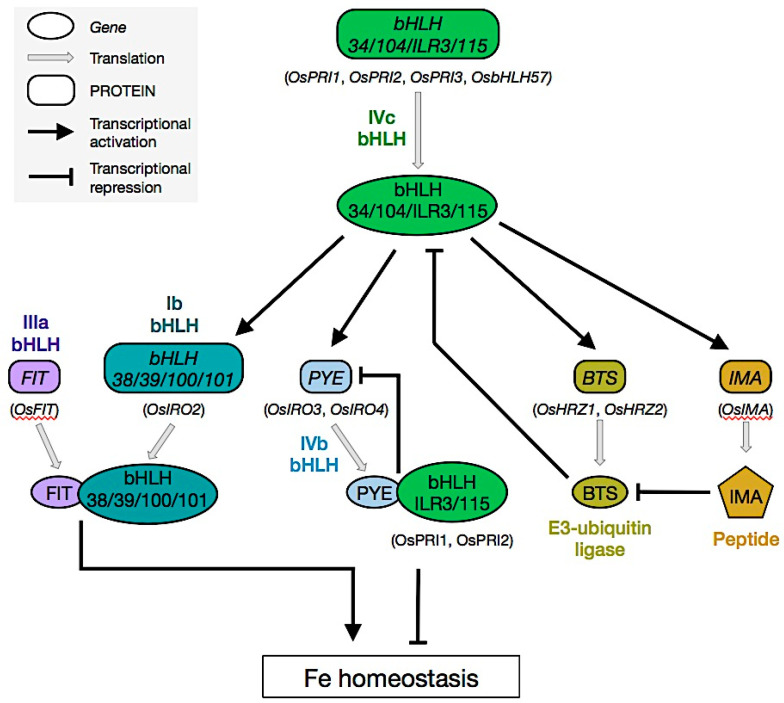
Transcriptional regulation of iron homeostasis: conserved mechanisms between grass and non-grass species. Upstream from the Fe homeostasis transcriptional regulatory network is clade IVc bHLH transcription factors (i.e., the Arabidopsis IDT1/bHLH34, bHLH104, ILR3/bHLH105 and bHLH115, and the rice OsPRI1/OsbHLH60, OsPRI2/bHLH58, OsPRI3/OsbHLH59 and bHLH57). Among the conserved target genes of clade IVc bHLH between grass and non-grass species are: (i) Clade Ib bHLHs (i.e., *bHLH38*, *bHLH39*, *bHLH100* and *bHLH101*, and the rice *OsIRO2*/*OsbHLH56*) whose encoded proteins interact with FIT/OsFIT to activate the expression of Fe uptake genes. (ii) Clade IVb bHLHs (the Arabidopsis *PYE*, and rice *OsIRO3* and *OsIRO4*) whose encoded proteins interact with clade IVc bHLH (i.e., the Arabidopsis ILR3/bHLH105 and bHLH115, and the rice OsPRI1 and OsPRI2) to negatively regulate Fe homeostasis and their own expression. These interactions allow balancing Fe uptake and Fe storage to meet the Fe demand of the plant for its growth and development. (iii) Hemerythrin E3-ubiquitin (E3-ubi) ligases (i.e., the Arabidopsis *BTS*, and the rice *HRZ1* and *HRZ2*) that are involved in the degradation of clade IVc bHLH transcription factors (i.e., the Arabidopsis ILR3/bHLH105 and bHLH115, and the rice OsPRI1 and OsPRI2). **(iv)**
*IMAs* that encode peptides that interact with and negatively modulate the hemerythrin E3-ubiquitin ligase activities (i.e., the Arabidopsis BTS, and the rice HRZ1 and HRZ2) that participate in the inter-organ signalling of cellular Fe status for fine-tuning Fe uptake in roots.

**Table 1 plants-12-00384-t001:** Transcription factors involved in the control of iron homeostasis in plants.

TF Family	Gene Name	Species
ABI3/VP1	IDEF1	*O. sativa*
ARF	OsARF12	*O. sativa*
ARF	OsARF16	*O. sativa*
bHLH	AtbHLH6 (MYC2)	*A. thaliana*
bHLH	AtbHLH11	*A. thaliana*
bHLH	AtbHLH18	*A. thaliana*
bHLH	AtbHLH19	*A. thaliana*
bHLH	AtbHLH20	*A. thaliana*
bHLH	AtbHLH25	*A. thaliana*
bHLH	AtbHLH29 (FIT)	*A. thaliana*
bHLH	AtbHLH34 (IDT1)	*A. thaliana*
bHLH	AtbHLH38	*A. thaliana*
bHLH	AtbHLH39	*A. thaliana*
bHLH	AtbHLH47 (PYE)	*A. thaliana*
bHLH	AtbHLH100	*A. thaliana*
bHLH	AtbHLH101	*A. thaliana*
bHLH	AtbHLH104	*A. thaliana*
bHLH	AtbHLH105 (ILR3)	*A. thaliana*
bHLH	AtbHLH115	*A. thaliana*
bHLH	AtbHLH121 (URI)	*A. thaliana*
bHLH	OsbHLH56 (OsIRO2)	*O. sativa*
bHLH	OsbHLH57	*O. sativa*
bHLH	OsbHLH58 (OsPRI2)	*O. sativa*
bHLH	OsbHLH59 (OsPRI3)	*O. sativa*
bHLH	OsbHLH60 (OsPRI1)	*O. sativa*
bHLH	OsbHLH63 (OsIRO3)	*O. sativa*
bHLH	OsbHLH133	*O. sativa*
bHLH	OsbHLH156 (OsFIT)	*O. sativa*
bHLH	GmbHLH57	*G. max*
bHLH	GmbHLH300	*G. max*
bHLH	FER	*S. lycopersicum*
bHLH	SlbHLH68	*S. lycopersicum*
bHLH	MxIRO2	*M. xiaojinensis*
bHLH	MxFIT	*M. xiaojinensis*
bHLH	PtFIT	*P. tremula*
bHLH	PtIRO	*P. tremula*
bHLH	MdbHLH18 (SAT1)	*M. domestica*
bHLH	MdbHLH104	*M. domestica*
bHLH	NtbHLH1	*N. tabacum*
bHLH	CmbHLH1	*C. morifolium*
bHLH	CmbHLH38 (FEFE)	*C. melo*
bHLH	GmORG3	*G. max*
bZIP	SlHY5	*S. lycopersicum*
B3	ABI3	*A. thaliana*
B3	FUS3	*A. thaliana*
B3	LEC2	*A. thaliana*
C2H2	ZAT12	*A. thaliana*
EIL	EIN3	*A. thaliana*
EIL	EIL1	*A. thaliana*
ERF	ERF4	*A. thaliana*
ERF	ERF72	*A. thaliana*
ERF	ERF95	*A. thaliana*
ERF	ERF96	*A. thaliana*
ERF	ERF109	*A. thaliana*
ERF	MxERF4	*M. xiaojinensis*
ERF	MbERF4	*M. baccata*
ERF	MbERF72	*M. baccata*
MYB (R2R3)	MYB10	*A. thaliana*
MYB (R2R3)	MYB28	*A. thaliana*
MYB (R2R3)	MYB29	*A. thaliana*
MYB (R2R3)	MYB72	*A. thaliana*
MYB (R2R3)	MdMYB58	*M. domestica*
MYB (R2R3)	MxMYB1	*M. xiaojinensis*
NAC	IDEF2	*O. sativa*
NF-YC	NF-YC1 (HAP5A)	*A. thaliana*
WRKY	WRKY12	*A. thaliana*
WRKY	WRKY46	*A. thaliana*
YABBY	INO	*A. thaliana*

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
