# Peer review of "Iron Nutrition in Plants: Towards a New Paradigm?"

_plants, 2023, doi:10.3390/plants12020384_

Round 1

Reviewer 1 Report

This review article entitled “Iron nutrition in plants: towards a new paradigm?by Lee et. al., reviewed recent findings on plant Fe uptake and the regulation process. Iron is a plant micronutrient. Fe availability influences crop yield, product quality, and human nutrition. Therefore, this review will compile recent knowledge to the researcher. In my view, this review is logically written and appropriately interpreted. However, I have few concerns that that need to be addressed prior to the publication.

Introduction:

Page 1, Line 30: Please provide more human health issues regarding Fe deficiency.

Page 5, Line 129: Please provide citations.

Conclusion:

Please trim the conclusion sections, seems too long.

Author Response

This review article entitled “Iron nutrition in plants: towards a new paradigm?” by Lee et. al., reviewed recent findings on plant Fe uptake and the regulation process. Iron is a plant micronutrient. Fe availability influences crop yield, product quality, and human nutrition. Therefore, this review will compile recent knowledge to the researcher. In my view, this review is logically written and appropriately interpreted. However, I have few concerns that that need to be addressed prior to the publication.

Introduction:

Page 1, Line 30: Please provide more human health issues regarding Fe deficiency.

We have modified the sentence accordingly.

Page 5, Line 129: Please provide citations.

We have included a reference.

Conclusion:

Please trim the conclusion sections, seems too long.

As suggested, we have shortened the conclusion.

Reviewer 2 Report

In my opinion submitted manuscript is an interesting in aspect of: 1) importance of iron for all  organisms, 2) molecular factors affecting the uptake of iron by two groups of plants. However, in the proces of Fe uptake very important factors are: 1) soil rection, especially in the conditions in which pH regulation to 6-7 is the main action in plant fertilization and better use of the main nutrients as well as better growth of plant roots and plants, 2) humus as an important element of soil which plays significant role in the Fe management in the alcaline soils.

Author Response

In my opinion submitted manuscript is an interesting in aspect of: 1) importance of iron for all  organisms, 2) molecular factors affecting the uptake of iron by two groups of plants.

However, in the process of Fe uptake very important factors are: 1) soil reaction, especially in the conditions in which pH regulation to 6-7 is the main action in plant fertilization and better use of the main nutrients as well as better growth of plant roots and plants, 2) humus as an important element of soil which plays significant role in the Fe management in the alkaline soils.

We agree and we have added this important point in the conclusion with a new reference.